# Feasibility of Genipin to Evaluate Chitosan Rainfastness for Biopesticide Applications

**DOI:** 10.3390/ijms26031031

**Published:** 2025-01-25

**Authors:** Solène Meynaud, Yunhui Wang, Gael Huet, Emmanuel Ibarboure, Christian Gardrat, Véronique Coma

**Affiliations:** Univ. Bordeaux, CNRS, Bordeaux INP, LCPO, UMR 5629, F-33600 Pessac, France; solene.meynaud@bordeaux-inp.fr (S.M.); yunhui.wang@bordeaux-inp.fr (Y.W.); gael.huet@bordeaux-inp.fr (G.H.); ibarbour@enscbp.fr (E.I.); christian.gardrat@u-bordeaux.fr (C.G.)

**Keywords:** chitosan, biopolymer, rainfastness, fluorescence microscopy, biopesticides

## Abstract

Chitosan’s effectiveness as an antimicrobial coating for biocontrol depends on its resistance to rain. Unfortunately, to the best of our knowledge, there is currently no satisfactory method for assessing this resistance, which means that field tests have to be carried out to evaluate it in situ, which are difficult to implement and therefore unsuitable for optimizing formulations. This article explores the use of genipin to detect residual chitosan on surfaces after simulated rain, using fluorescence microscopy. A first study on real vine leaves using MacroFluo microscopy was carried out but showed limitations for the intended application, notably due to the requirement for high chitosan concentrations to achieve detectable signals. A semi-quantitative method based on confocal laser scanning microscopy was then developed on model leaves, as real leaves were unsuitable due to their autofluorescence. Among the tested models, Parafilm^®^ proved to be the most effective, showing sufficient fluorescence after reaction with genipin, even at low chitosan concentrations. For the first time, a method that does not require chromophore grafting onto chitosan has been proposed, allowing for the comparison of chitosan solution rainfastness under laboratory conditions. As an application, the effect of the counter ion on chitosan’s rain resistance was evaluated.

## 1. Introduction

Chitosan is a well-known biobased polysaccharide composed of randomly distributed N-glucosamine and N-acetylglucosamine units. It has gained significant attention as an alternative bioactive compound in agriculture owing to its antimicrobial and eliciting properties [1,2]. New regulations, such as the Ecophyto Plan in France [3], intended to reduce the use of phytopharmaceutical products and substitute them with safer active formulations. The prevailing context supports the adoption of biopesticides that rely on renewable antimicrobial agents, among which, chitosan stands out [4].

The efficacy of bioactive polymers, such as chitosan, in agriculture is greatly influenced by the persistence of the polymer on leaves. Antimicrobial activities observed in laboratories and greenhouses often surpass those in open fields, with rainfastness being one of the suspected factors contributing to this difference [5,6], as also observed with certain conventional treatments [7]. Hence, it is increasingly crucial to establish a method for assessing chitosan’s resistance on the leaf surface following its application. Rainfastness testing plays a significant role in enabling chitosan and chitosan-based solutions to emerge as a novel approach for disease treatment. These tests are highly versatile, offering the option to detect chitosan residues through two distinct media: the washing solution or the washed surface. In this context, surface detection is particularly advantageous, given that previous studies have already demonstrated the utility of techniques such as fluorescence labeling to detect chitosan after deposition on surfaces. However, it is important to emphasize that while the pre-deposition modification of the polymer is effective, this approach may not accurately reflect the raw product’s resistance. Factors such as the presence of ions and small oligomers may significantly influence rainfastness behavior. A more representative approach involves the direct modification of residual chitosan on the surface. Although no such methods have been reported in the literature to date, certain reagents, such as NaNO_2_ combined with 2,4-dinitrosalicylic acid [8] and Lugol’s solution [9], have shown the potential to induce fluorescence through reactions with chitosan. However, these reagents also react with other polysaccharides present in complex formulations, which limits their specificity for chitosan and reduces their utility in targeted applications. Genipin, obtained after the hydrolysis of geniposide extracted from Gardenia fruits, undergoes cross-linking reactions with the primary amine groups of chitosan to form fluorescent compounds [10,11]. These reactions involve the nucleophilic attack of genipin on the primary amine, resulting in the formation of an aldehyde. This is followed by the cyclization and nucleophilic substitution of the ester group, ultimately yielding a fluorescent product [12]. A reaction under alkaline conditions also yields side products resulting from the self-polymerization of genipin [13] via ring-opening polymerization, which decreases yield and light intensity.

The reactivity of genipin with amines has attracted significant attention, particularly after its ability to target amino acids in fingerprint residues was demonstrated, highlighting its potential for surface amine detection [14]. The cross-linking of chitosan with genipin has been studied for use in various applications, including improving the properties of wood [15,16], the membrane characterization of microcapsules [17], and developing novel biomaterials for tissue engineering [18]. However, to the best of our knowledge, this reaction has not been reported for the detection and quantification of native chitosan directly on surfaces. Moreover, its potential for evaluating the rainfastness of chitosan, by evaluating fluorescence loss in lab-controlled conditions following rain exposure, remains unexplored.

In this context, the present work aims to study a method for detecting and potentially quantifying residual chitosan on grapevine leaves or model leaf surfaces for the evaluation of chitosan rainfastness through the generation of fluorescent cross-linked chitosan. This method is much more dedicated to comparing the rainfastness of different chitosan-based formulations at a laboratory scale, facilitating the selection of the most effective formulations. Following the characterization of the selected chitosan, the feasibility of quantifying residual chitosan on vine leaves was initially evaluated using MacroFluo microscopy, as this technique enables the analysis of real plant material. However, due to some limitations encountered with this approach, the study was subsequently conducted on model leaves, and thus by using confocal laser scanning microscopy (CLSM), a more straightforward and available technique. In this study, the method was employed to investigate the influence of acetic and citric acids as counter ions on the rain resistance properties of chitosan.

## 2. Results

### 2.1. Chitosan Characterization

The chitosan was characterized in terms of deacetylation degree, molecular weight, XPS, and elemental analysis, and the results are presented on Table 1 (Appendix A).

Traces of Ca, Si, and Cl could have been due to the processing of crustacean exoskeleton components of the source material of chitosan [19].

### 2.2. Feasibility of the Evaluation of the Residual Chitosan Content on a Vine Leaf Using MacroFluo Microscopy

Because chitosan has the potential to control grapevine phytopathogens such as mildew or oidium, vine leaves were selected for the experiments with MacroFluo microscopy [20]. Experiments were performed in triplicate due to the heterogeneity of the fresh material. First, the calibration curve of a series of chitosan solutions in aqueous citric acid with known concentrations was elaborated. Microscopic images of genipin-treated chitosan spots were recorded (Figure 1 and Appendix A).

Fluorescence was not visible across all concentrations, and background fluorescence from the leaf surface was also observed. The mean intensity and area of the spots were then measured, and the relationships between fluorescence and chitosan concentrations were studied (Figure 2).

The concentration of the deposits did not appear to significantly influence the spot area, suggesting the potential feasibility of quantification. However, the relationship between fluorescence intensity and concentration was non-linear, with notable variations in intensity observed even at identical concentrations. Thus, the low concentrations involved made the establishment of a reliable calibration curve challenging, primarily due to difficulties in sample visualization. To address these issues, modifications to the experimental protocol were implemented such as in the concentration range and exposure parameters in order to improve the contrast and achieve a more uniform distribution of chitosan. As a result, microscopy images showed the elimination of halos and enhanced contrast between the sample and the background fluorescence of the leaf surface (Figure 3). Based on these optimizations, a calibration curve was established within 1 and 10 mg/mL.

No significant visual discrepancies were observed in the results with the change in exposure and gain. Therefore, 500 ms of exposure and gain 9 were retained as parameters for calibration (Appendix A).

The mean fluorescence intensity and area were then measured (Figure 4).

Linearity improved with higher concentrations, although considerable standard deviations were still evident. A simple test was performed using the initial concentration range of chitosan (1 mg/mL) (Figure 5) and the washed spot was barely visible and inhomogeneous, just like the unwashed spot. For the remainder of the study, model leaves were used with the CLSM technique instead of MacroFluo. Given the switch from real leaves to model leaves, it was no longer necessary to continue with the MacroFluo technique as CLSM offers several advantages, including greater simplicity, ease of access, and enhanced sensitivity, particularly when working with low chitosan concentrations.

### 2.3. Feasibility of Chitosan Evaluation on a Leaf Model Using CLSM

Models of leaves were employed to develop the method, owing to the complex interactions between genipin and natural compounds present on the surface of actual leaves. These interactions led to a lack of contrast between the background and the sample, making it challenging to obtain clear and distinguishable results. As shown in Figure 6, no clear differences were observed between the autofluorescence of leaves and the genipin-treated chitosan sample.

As a result, leaf models were used to create representative, malleable, and unreactive hydrophobic surfaces.

#### 2.3.1. Selection of an Appropriate Leaf Model

Three model leaves were created: carnauba wax, Parafilm^®^, and silicon hydrophobic matrices. To select the most appropriate model, the hydrophobicity and SFE were evaluated using contact angle measurements (Figure 7).

Color modification was observed on the carnauba wax model after contact with water, perhaps due to the migration of Tween 60, used as a plasticizer, from the wax to the water drop (Appendix A). The properties of Parafilm^®^ are close to those of real leaves; here, it was selected for use in the rest of the study. Although the surface topology of real leaves was not represented by this material, unlike carnauba wax, its hydrophobic characteristics aligned more closely with those of natural leaf surfaces, justifying its selection for the experimental work.

#### 2.3.2. Relationship Between Concentration and Fluorescence of Genipin-Treated Chitosan

Preliminary tests revealed that the intensity increased with higher concentrations of chitosan, even in cases where the typical cross-linking browning was not visibly visible to the naked eye (Figure 8).

The relationship between chitosan concentration and fluorescence intensity was first studied. Chitosan solutions with known concentrations in citric acid were deposited on Parafilm^®^ and the intensity of fluorescence was evaluated after the genipin treatment. The microscopic images showed that, even at low concentrations, the fluorescence intensity was sufficient to be well observed, and the presence of chitosan could be detected (Figure 9 and Appendix A).

In addition, fluorescence was emitted only from genipin-treated chitosan spots, which could be easily distinguished from the background (residual genipin, other impurities). By defining some areas of interest on the images, the software was able to record and calculate the intensities of the fluorescence, and it was then possible to have the mean of the intensities and the areas of the spots. The relationships between the mean intensity of fluorescence vs. the initial chitosan concentration in the formulation were then studied (Figure 10).

The mean intensity vs. chitosan concentration seemed to follow a linear relationship, without passing through the origin. The reaction with genipin created a network due to the cross-linking of chitosan, intensifying the three-dimensional aspect of the deposition spot and its irregularity, giving higher fluorescence intensities than expected, which could explain why the calibration curve did not pass through the origin. Moreover, this calibration seems acceptable because the size of the deposits at different concentrations remained constant (Figure 11).

#### 2.3.3. Evaluation of the Residual Chitosan After Rainfastness Test

Chitosan solutions, previously deposited on Parafilm^®^ as before, were submitted to the rain simulation step. After the genipin reaction, all spots were visible by CLSM (Figure 12).

The mean fluorescence intensities were then measured before and after the rain simulation step. Using the mean intensities, the percentage of leached chitosan could be estimated at 85%, going from 159 to 24 for the fluorescence intensity. To investigate whether the counter ion could influence the rainfastness of chitosan formulations, the behavior of the chitosan–citric acid spot was compared to that of the chitosan–acetic acid spot. The required concentration to solubilize chitosan in this study was 3 and 0.3% (*w*/*v*) for citric acid and acetic acid, respectively. The CLSM images before and after the rain simulation are given in Figure 13.

By the calculation of the loss of fluorescent intensity, the rainfastness was estimated to be 35%. Unfortunately, the intensity loss was not only related to chitosan rainfastness but also to spreading, proved by the increase in the spot surface, indicating that the calculated intensity losses cannot be correlated with chitosan loss.

## 3. Discussion

As previously mentioned, one of the key advantages of using genipin to assess rainfastness is its ability to analyze the leaf surface directly without the need for chitosan derivatization prior to the deposit. In addition, the reaction between genipin and chitosan can take place in ethanol, a solvent that offers the benefit of preserving the insolubility of chitosan. The analysis of the resulting fluorescence, conducted using MacroFluo or CLSM, is discussed below.

### 3.1. Feasibility of the Evaluation of the Residual Chitosan Content on a Vine Leaf Using MacroFluo Microscopy

MacroFluo microscopy uses the fluorescence of incident light to differentiate leaf autofluorescence from sample fluorescence, while simultaneously capturing all the fluorescence emitted from the surface. This technique effectively overcomes the challenge of distinguishing between intrinsic autofluorescence and the sample-specific fluorescence by providing a complete visualization of the emitted fluorescence.

This method may be more suitable for relative quantification, considering that the deposition areas remained relatively constant over the selected concentration range. However, it should be noted that achieving good image quality required the use of very high concentrations of chitosan, which raises questions about the possibility of employing MacroFluo microscopy for this method.

The preliminary rainfastness test showed that it was difficult to see the chitosan deposited at 1 mg/mL before the rain simulation, which highlighted the limitations of the method. This technique could still be used to detect chitosan residues from cropland samples after rain episodes if chitosan concentrations are high enough to have a high fluorescence for the “before” reference.

Another technique was therefore studied, still based on the detection of chitosan by the genipin reaction, but with the measurement of fluorescence by confocal microscopy.

### 3.2. Feasibility of Chitosan Evaluation on a Leaf Model Using CLSM

CLSM is a type of fluorescence microscopy that uses out-of-focus light rejection to guarantee better resolution while remaining quick and easy to use. CLSM analysis has already been used to qualitatively detect chitosan cross-linked with genipin [17].

CLSM has been widely employed for various quantification purposes in scientific research. For instance, it has been utilized to quantify polymer distribution by measuring coating intensity and thickness [21,22]. CLSM has also been employed to assess rainfastness by quantifying fluorescent surface area [23]. However, it should be noted that CLSM is not generally used for comprehensive quantitative studies based solely on intensity changes. This limitation is due to its focus on a single plane, which does not take into account the overall fluorescence of the sample. The possibility of a relative quantitative analysis was nevertheless experimented with in this work in order to be able to compare chitosan formulations, and for this purpose, CLSM is a technique that offers the advantage of being rapid and simple to implement.

Hydrophobicity plays an important role as it determines the interaction between water and the leaf surface, influencing the spreading behavior of chitosan solution by the retention during pulverization [24,25] but also its adhesion after drying [26]. The leaf hydrophobic surface, called the cuticle, is mainly composed of cutin, a hydroxy fatty acid and glycerol polyester [27,28], and cuticular waxes [29]. The composition of the cuticle depends on the species but also on the age, location, and climatic conditions, which influence the properties, in particular, the wettability and permeability [30,31,32]. The formation of three-dimensional crystals or the presence of trichomes on the surface of the cuticle is also a parameter that influences the hydrophobicity of the leaf [31,33].

For rain resistance tests, it is necessary to mimic conditions that are as close as possible to field conditions, which is why a study of different leaf models was carried out. The selection of models was based on the hydrophobic character and the free energy (SFE) of the surface. These parameters had to be as close as possible between the real leaf and the model leaf. The leaf models studied here have different hydrophobicity and SFE. Carnauba is mainly composed of long chain waxy esters and fatty acids [34]. According to suppliers, Parafilm^®^ is a blend of paraffin waxes and polyolefins, and finally, silicone owes its hydrophobic properties to the presence of silicone elastomers.

Significant differences in SFE were obtained for the leaf models, indicating a possible difference in the adhesion of the chitosan solutions. As the leaf model should not interfere with the physical and chemical properties of the chitosan solution, the carnauba wax model was not selected for the rest of the study. However, this model could still be valuable for evaluating rainfastness on hydrophobic systems after improving the formulation to prevent the solubilization of chemicals in water droplets. Furthermore, it is the only model that simulates leaf topography, another important factor for adhesion and spreading. But, as for the silicone model, it exhibited significant deviation in surface free energy (SFE) from that observed in real leaves, as initially tested (Appendix A). Parafilm^®^, as a simple and cost-effective model, was finally selected for this study, in agreement with some previously reported work [35,36]. However, Parafilm^®^ is not suitable for simulating specific vegetal species, as real leaves exhibit highly variable surface properties and topographies that depend on factors such as species and leaf age [37,38]. Thus, while Parafilm^®^ serves as a general model for assessing broad properties, it lacks the specificity required for more detailed, species-targeted simulations.

The first limitations were observed when establishing the calibration curve, with variations in the surface and three-dimensional appearance of the drop. This resulted in the calibration curve not passing through the origin. Additionally, droplets from the rain-wash test showed spreading when compared to the initial deposits, suggesting that the lixiviation percentages obtained with the fluorescence decrease are therefore not representative and cannot be used for a quantitative analysis of rainfastness. However, this technique remains useful for evaluating the improvement in rainfastness by modifying the formulation, as the impact of chitosan counter ions on rainfastness is established. It is well established that the choice of chitosan counter ions, which are used to solubilize chitosan in aqueous media, influences the water solubility of the final chitosan film [39]. For this, two organic acids commonly used with chitosan were tested as counter ions. Chitosan exhibits lower solubility in citric acid compared to acetic acid [40], which necessitated a higher concentration of citric acid (3% *w*/*v*) versus acetic acid (0.3% *w*/*v*) to achieve solubilization in this study (*w*/*v)*. Clearly, lower fluorescence loss was observed with chitosan pre-solubilized in acetic acid compared to that in citric acid, despite a slight variation in spreading behavior, thus showing the importance of this parameter in the rain resistance of chitosan-based formulations. A more comprehensive study of various counter ions could thus be carried out to address this significant limitation in the application of chitosan for bioactive agricultural purposes.

The CLSM method offers a viable alternative for assessing the simulated rain resistance of chitosan, in contrast to techniques that require chemical derivatization. A major drawback of previously reported chitosan detection methods, such as fluorescein isothiocyanate labeling [23], is the potential risk of misinterpretation due to interferences. This interference arises from the possible involvement of the labeling in the interactions between chitosan and plant leaf surfaces, or between chitosan molecules when grafted with a bulky and hydrophobic compound. Thus, the use of post-deposition modification with genipin is advantageous for progressing towards a more representative visualization of the rain resistance of chitosan during the creation of biocontrol formulations, even though the use of a model leaf is mandatory. It is important to remember that CLSM focuses precisely on a single plane, not providing the entire fluorescence. This complicates the calculation of the total emission intensity in some cases, for example, when establishing a 3D model by stacking planes, which could significantly reduce the relevance of this method for the intended application. In the end, this method can be employed for a semi-quantitative evaluation of chitosan retention under two scenarios: (1) when the surfaces of the spots are similar before and after rain simulation, the decrease in fluorescence intensity can be correlated to the loss of chitosan; and (2) when the fluorescence intensity remains comparable before and after rain simulation, the reduction in the surface area can be correlated with the loss of chitosan.

Moreover, the genipin-based method could be used to assess the number of wash steps required to achieve complete fluorescence disappearance, providing a measure of chitosan removal.

## 4. Materials and Methods

### 4.1. Materials

Chitosan obtained from crab shells was purchased from SPN Agrobio^®^ (20 cps SPN, Varades, France). Citric acid, glacial acetic acid and carnauba wax from Sigma Aldrich (St. Louis, MO, USA), and Tween 60 and genipin from Fisher Scientific (Loughborough, UK). A silicone kit (Bluesil RTV 3428 white kit) was purchased from Elkem (Saint-Fons, France). Parafilm^® “^M” was bought from VWR (Fontenay-sous-Bois, France). Genipin solution was prepared in 99% ethanol and stored in an ambered glass bottle at 4 °C. Deionized water was prepared using the MilliQ system from Merck (Taufkirchen, Germany). All chemicals were used as received, without further purification.

Vine leaves (*Vitis vinifera*, Fer Servadou) were collected from a local organic garden, washed, dried, and cut to a proper size and fixed on a microscope slide with adhesive tape to reduce the deformation of the leaf surface during drying.

### 4.2. Methods

#### 4.2.1. Chitosan Characterization

Deacetylation degree

The deacetylation degree (DD) of the chitosan was determined by ^1^H NMR (Liquid-state 400 MHz, Bruker Advance, Ettlingen, Germany). Chitosan was solubilized in D_2_O by adding DCl (1% *v*/*v*) under magnetic stirring overnight. The ^1^H NMR spectrum was recorded at 80 °C in order to shift the water signal.

Molecular weight

The chitosan size was evaluated by Size Exclusion Chromatography (SEC) analysis (Ultimate 3000 system from Thermoscientific, Waltham, MA, USA). The sample was solubilized in a sodium acetic acid/acetate buffer solution (acetic acid 0.3 M/sodium acetate 0.2 M) at the concentration of 5 mg/mL. The solution was filtrated by a cellulose acetate syringe filter (0.23 µm) before being injected into two combined SEC columns, Tosoh PWXL G3000 and G4000 (7.8 mm × 300 mm), with exclusion limits from 5 kDa to 300 kDa. The flow rate was fixed at 0.6 mL/min and the column temperature at 25 °C. The absolute method by increment in the refractive index was applied to calculate the molecular weight of chitosan. The refractive indexes of five chitosan solutions going from 5 to 20 mg/mL were measured and the dn/dc value for molecular weight calibration was automatically calculated by Astra software (version 7.3.2.21). Three detectors were used, which were a multi-angle light scattering (MALS) detector, a diode array detector (UV), and a differential refractive index detector (dRI) from Wyatt Technology (Santa Barbara, CA, USA).

Surface analysis by XPS

A ThermoFisher Scientific K-ALPHA spectrometer was used for XPS surface analysis with a monochromatized Al-Kα source (hν = 1486.6 eV) and a 400 μm X-ray spot size. Powders were pressed onto indium foils. The full spectra (0-1100 eV) were obtained with a constant pass energy of 200 eV, while high-resolution spectra were recorded with a constant pass energy of 40 eV. Charge neutralization was applied during the analysis. High-resolution spectra were quantified using the Avantage software (version v4) provided by ThermoFisher Scientific.

Elemental analysis (C, H, N, O)

Elemental analysis was performed by SGS (Evry-Courcouronnes, France).

#### 4.2.2. Chitosan Solution Preparation

A chitosan solution (10 mg/mL) was prepared in aqueous either acetic or citric acid (0.29 and 3.33% *w*/*v*, respectively) by vigorous magnetic stirring overnight to lead to total dissolution. This chitosan solution was then diluted with deionized water to desired concentrations, namely 8, 5, 2, 1, 0.8, 0.5, 0.2, 0.1, and 0.05 mg/mL.

#### 4.2.3. MacroFluo Analyses

All MacroFluo microscopy analyses were performed with a Leica Z16 APO with a color CCD DFC300FX camera and a grayscale CCD DFC350 FX camera (Leica Microsystems, Wetzlar, Germany) under channel 3 (620–750 nm) for the fluorescence.

Calibration on vine leaf

Chitosan solutions in citric acid were spotted (2 µL) on a vine leaf previously fixed on a microscope glass slide and reacted with genipin as follows.

The solution spots were dried at 40 °C for 24 h. Then, genipin solution in ethanol (0.2%, *w*/*v*) was dropped on the dried chitosan spots, 5 µL for each spot. The spots were thermally treated at 40 °C for 6 h. The genipin deposition was repeated and the spots were again treated at 40 °C for 24 h. Genipin deposition was repeated in the same way and treated this time at 40 °C for 48 h. This experiment was realized in triplicate.

Chitosan spots were then observed by MacroFluo microscopy under normal light (bright field) and under channel 3 with various exposition, fusion, and gain values to achieve better contrast between the leaf and genipin-treated chitosan.

Images were treated with the ImageJ software (version 1.54g, open source). The intensity of the sample was calculated automatically by the software using the free drawing setting, and the mean intensity of fluorescence was obtained after a subtraction of the intensity of background (leaf fluorescence). The area was calculated automatically by the software, using the free drawing setting. Then, the mean intensity–concentration and drop-off surface–concentration were fitted with cloud points in Microsoft Excel.

Simulated rain resistance test on vine leaf

Simulated rain resistance was assayed using a method already described [23,41]. Briefly, 2 drops (0.4 µL each) of 1 mg/mL chitosan solutions in citric acid were spotted on a vine leaf previously fixed on a microscope glass slide with a microsyringe (Hamilton, Bonaduz, Switzerland), with a significant distance between both. The spots were dried in an oven at 40 °C for 24 h. Then, one of the two spots was washed dropwise (about 1 drop/s) with 1 mL of deionized water with a syringe, with an angle around 45°, from about 1 cm height. This test was carried out to simulate rain washing in the field; in other words, it was a rainfastness test of the chitosan sample. After drying for 2 h at 40 °C, the spots were treated as follows: Genipin solution in ethanol (0.2%, *w*/*v*) was dropped on the dried chitosan spots, with 5 µL for each spot, and slides were put in an oven at 40 °C for 24 h. This reaction step was repeated for each spot to be treated further, at the same temperature, but for at least 72 h to promote the cross-linking reaction between chitosan and genipin. Reacted spots were then submitted to MacroFluo microscopy (Leica Microsystems, Wetzlar, Germany).

#### 4.2.4. Elaboration of Leaf Models

Three leaf models were studied: (1) a carnauba wax-based model, (2) a silicone model, and (3) the Parafilm^®^.

Carnauba wax model

The carnauba wax model was fabricated in three steps (Figure 14).

Step A: A negative-mold created with a silicone kit according to the specification of the provider was poured into a polystyrene Petri dish. The viscous fluid was degassed in a desiccator under vacuum for 1 h. Then, a grapevine leaf was slightly pressed onto the silicone. The silicone was cured at room temperature for 24 h, the leaf was removed, and the mold was cut into the desired shape.

Step B: Similarly to pre-existing plasticization processes [42], carnauba wax was melted at 110 °C, and Tween 60 (polysorbate), acting as a plasticizer, was added (carnauba/Tween ratio of 5/3 *w*/*w*). Then, the mixture was mechanically mixed at 5000 rpm for 15 min at 110 °C and slowly cooled down to room temperature.

Step C: A small amount of the plasticized carnauba wax was placed in the middle of a microscope slide and melted at 105 °C in a vacuum oven. The silicone mold was placed onto the melted wax to “print” the leaf surface. Finally, the system was degassed in a vacuum oven at 105 °C for at least 15 min. When wax was sufficiently solidified after cooling to 60–70 °C, the silicone mold was carefully removed, and the wax leaf model on the microscope slide was cooled slowly in the oven to avoid cracking.

Silicone model

The silicone model was made by putting the silicone (same as before in Section 4.2.4 “Carnauba wax model”) between two microscope slides. To obtain a smooth surface on both slides, a slow horizontal movement to separate the two silicone covered slides was made.

Parafilm^®^ model

The Parafilm^®^ model was simply made by putting Parafilm^®^ on a microscope slide.

#### 4.2.5. Surface Analysis of Leaf Models

Surface analysis was performed using the sessile drop method (Goniometer Krüss DSA 100). Surface free energy (SFE) was calculated using the OWRK method [43,44] with water, diiodomethane, and ethylene glycol as solvents (2 μL of water and ethylene glycol and 1.3 μL of diiodomethane; 5 records for each).

#### 4.2.6. Confocal Analyses

All CLSM images were acquired on an inverted Leica TCS SP5 microscope equipped with an HCX PL APO CS 10× DRY objective in fluorescence mode. The laser outputs were controlled via the Acousto-Optical Tunable Filter (AOTF) and the collection window using the Acousto-Optical Beam Splitter (AOBS) and photomultipliers (PMTs). Samples were excited with a laser at 561 nm (21%) and measured with emission settings at 565–650 nm. The Helium–Neon laser at 633 nm (10%) was only used in transmission mode. Images were collected in simultaneous mode using a rate at 400 Hz and a resolution of 512 × 512 pixels. 

Feasibility test on various chitosan concentrations

The chitosan solutions in citric acid were dropped on filter paper (Whatman 1, diameter 110 mm, 0.1 mL for each concentration) before drying overnight under a fume hood at room temperature. Then, genipin solution (0.2%, *v/v* ethanol) was thoroughly sprayed on the filter paper and dried overnight at room temperature; the spray-drying procedure was repeated once. After two cycles of spraying and drying, the filter paper was heated at 45 °C for 2 days. Finally, the area with samples on the paper were cut down with scissors and submitted to CLSM.

Calibration

Chitosan solutions at 0.1, 0.2, 0.5, 0.8, and 1 mg/mL in citric acid were spotted on the Parafilm^®^ model with a microsyringe (Hamilton, 0.4 µL for each concentration). The solution spots were dried at 40 °C for 24 h. After drying, chitosan deposits were reacted with genipin as described in Section 4.2.3 for the simulated rain resistance test on vine leaf. Every sample was then analyzed through CLSM at an excitation wavelength of 561 nm. The focal plane of the analysis was set to the most intense and homogeneous fluorescent surface. The experimental conditions were optimized starting from the highest concentration by adjusting both the photomultiplier gain and the laser intensity to avoid saturation and allow the lowest concentrations to be detected.

Images were treated with the ImageJ software (open sourced). Intensity was calculated automatically by the software as a mean of the pixels’ intensity of the entire fluorescence surface through the free drawing setting. The area was also calculated automatically by the software, using the free drawing setting from the transmission mode image of the spots. Then, the intensity–concentration and drop-off surface–concentration were fitted with cloud points in Microsoft Excel. 

Simulated rain resistance test

Simulated rain resistance was assayed as described in Section 4.2.3 for the simulated rain resistance test on vine leaf with chitosan at 1 mg/mL either in citric or acetic acid. The washed and unwashed spots were treated as mentioned in Section 4.2.3 for the simulated rain resistance test on vine leaf and submitted to CLSM. The fluorescence intensity and areas of unwashed and washed spots were evaluated by ImageJ software.

## Figures and Tables

**Figure 1 ijms-26-01031-f001:**
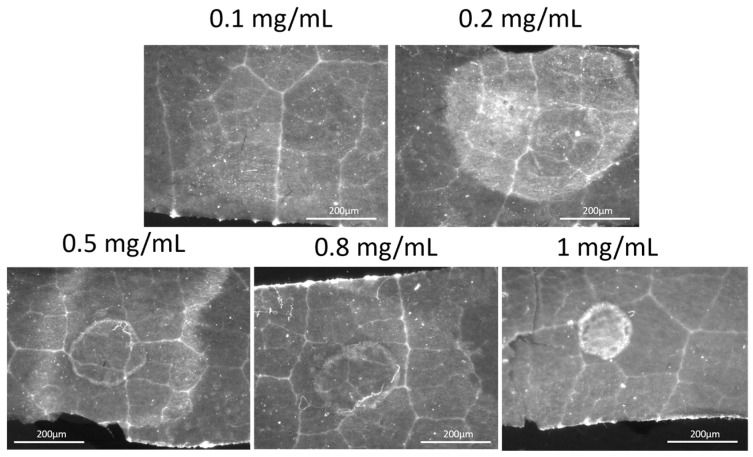
MacroFluo microscopy of genipin-treated chitosan spots (chitosan concentrations from 0.1 to 1 mg/mL) under fluorescence in red channel; exposition of 250 ms.

**Figure 2 ijms-26-01031-f002:**
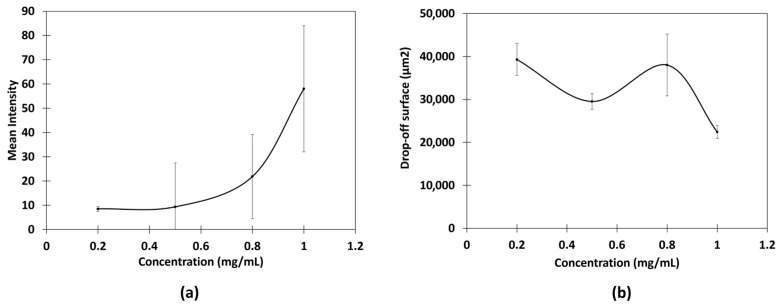
Relationships between (**a**) mean fluorescent intensity and chitosan concentration and (**b**) drop-off surface and chitosan concentration in citric acid. Means from triplicate measurements.

**Figure 3 ijms-26-01031-f003:**
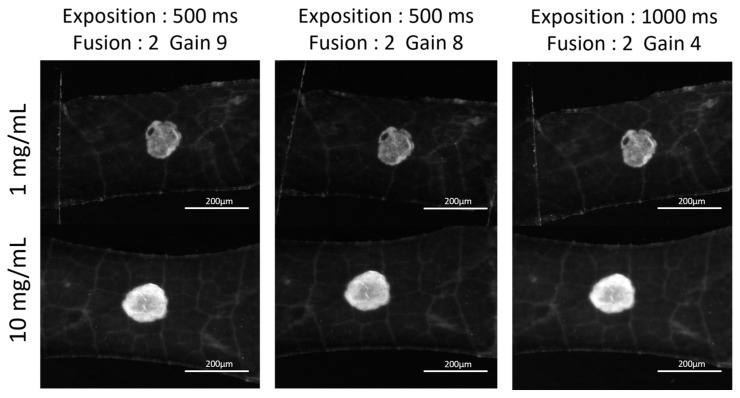
MacroFluo microscopy of genipin-treated chitosan spots under fluorescence in red channel with different settings.

**Figure 4 ijms-26-01031-f004:**
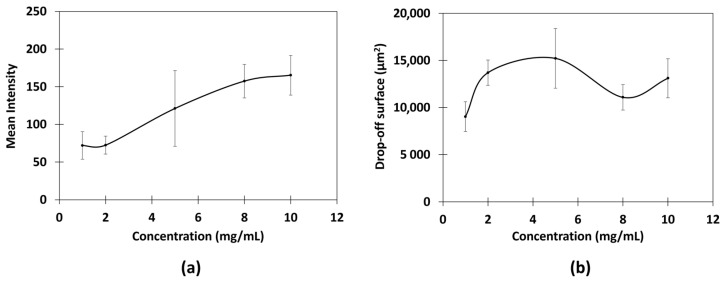
Relationships between (**a**) mean intensity and concentration and (**b**) drop-off surface and concentration of chitosan in citric acid; 500 ms of exposure, fusion 2, and gain 9; triplicate measurements.

**Figure 5 ijms-26-01031-f005:**
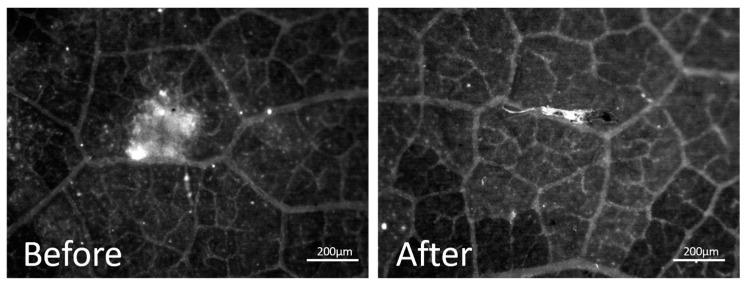
MacroFluo microscopy before and after rainfastness test on chitosan spot.

**Figure 6 ijms-26-01031-f006:**
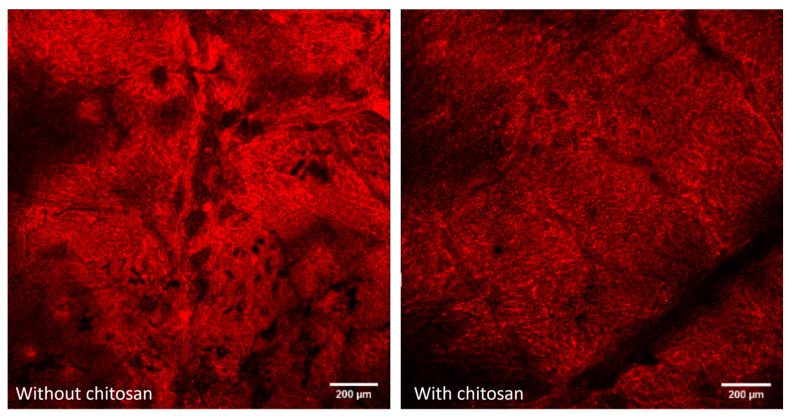
CLSM of fresh vine leaf without and with chitosan droplet at 1 mg/mL on fresh vine leaf treated with genipin.

**Figure 7 ijms-26-01031-f007:**
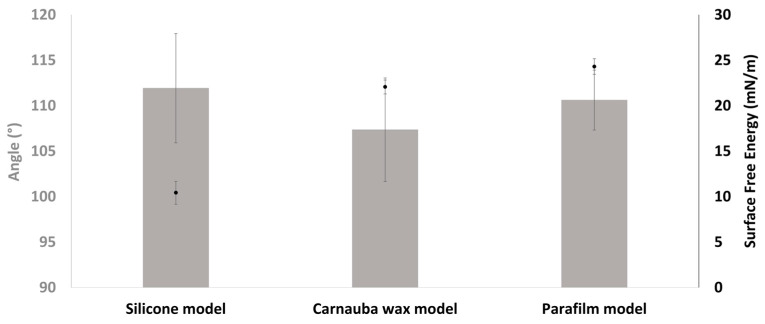
Water contact angle (gray) and surface free energy (black) of leaf models.

**Figure 8 ijms-26-01031-f008:**
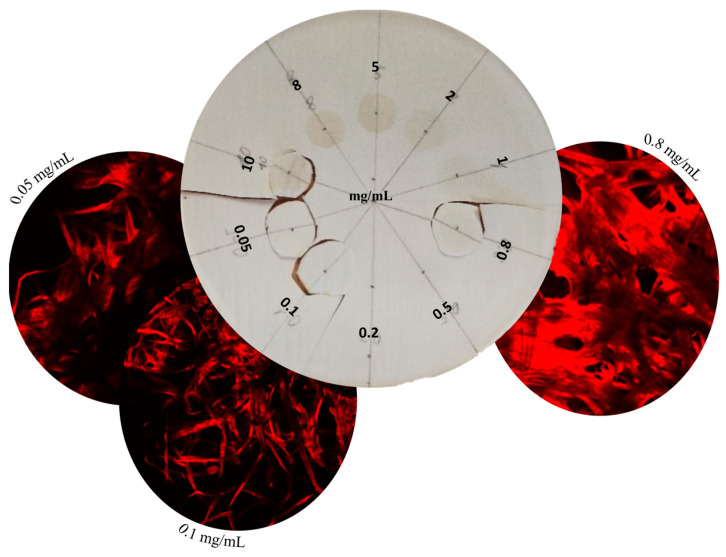
Genipin-treated chitosan on filter paper at different chitosan concentrations from 0.05 to 10 mg/mL and confocal laser scanning microscopy images.

**Figure 9 ijms-26-01031-f009:**
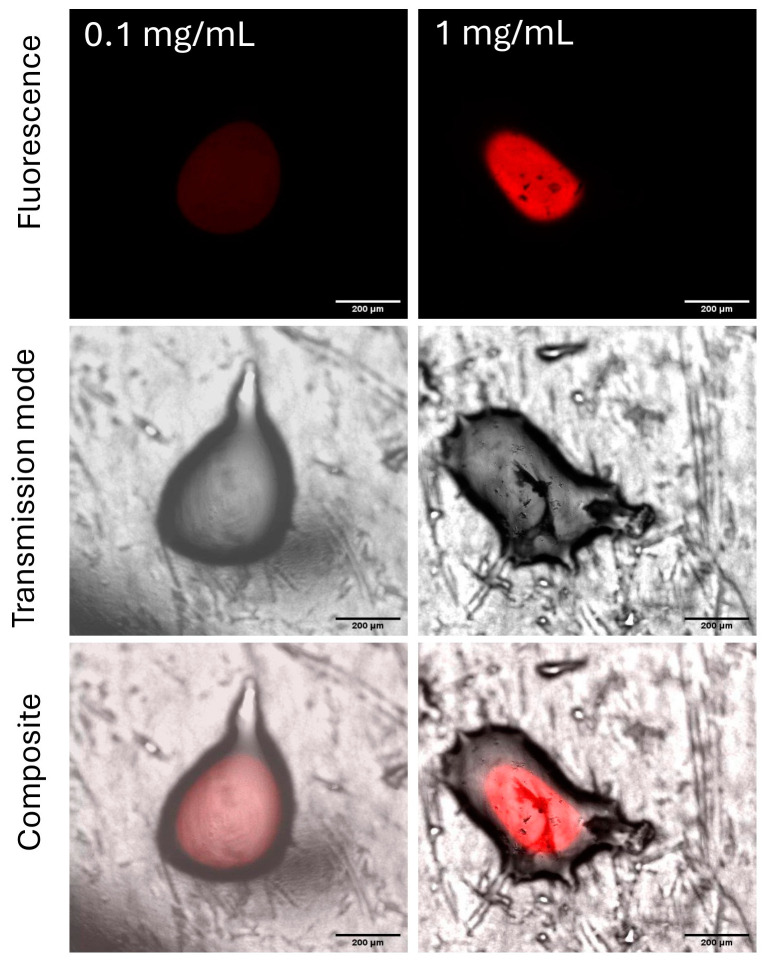
CLSM in fluorescence and transmission mode and composite image of chitosan in citric acid.

**Figure 10 ijms-26-01031-f010:**
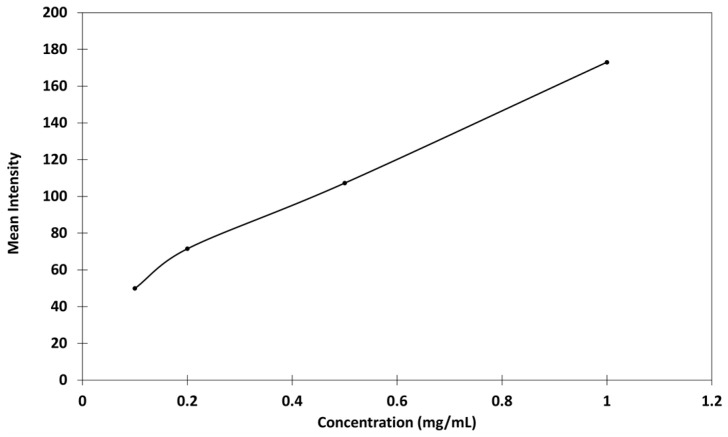
Mean intensity–concentration relationship for genipin-treated chitosan in citric acid, on Parafilm^®^ as leaf model.

**Figure 11 ijms-26-01031-f011:**
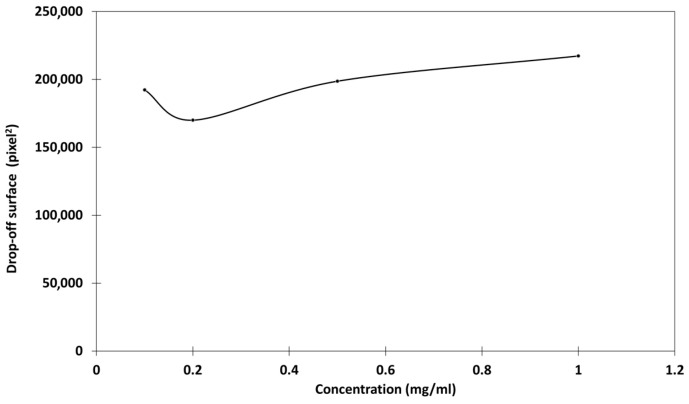
Relationship between the drop-off surface and the concentration of chitosan in citric acid on Parafilm^®^ as the leaf model. The deposit sizes were measured after the genipin reaction.

**Figure 12 ijms-26-01031-f012:**
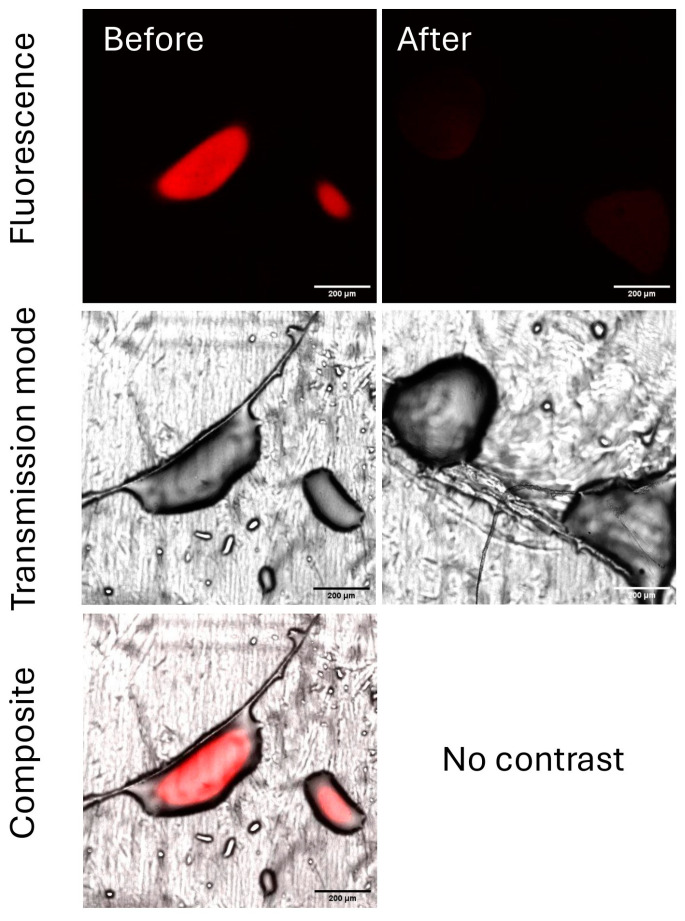
CLSM images of chitosan in citric acid before and after the rain simulation step.

**Figure 13 ijms-26-01031-f013:**
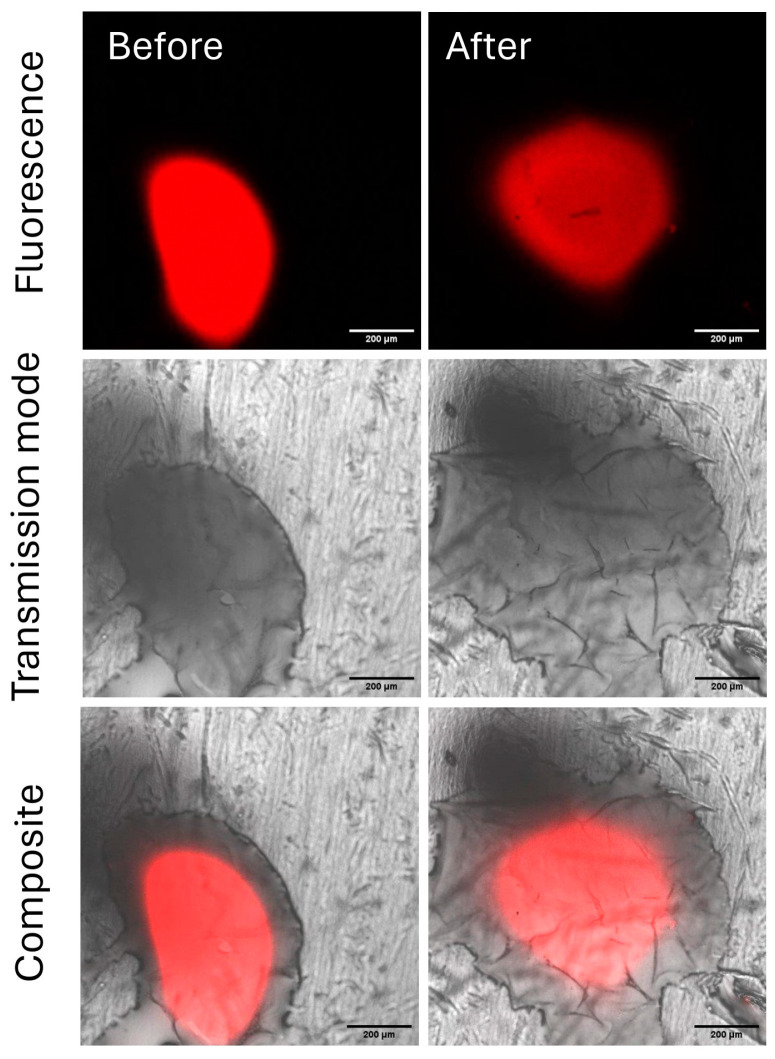
CLSM images of chitosan in acetic acid before and after the rain simulation step.

**Figure 14 ijms-26-01031-f014:**
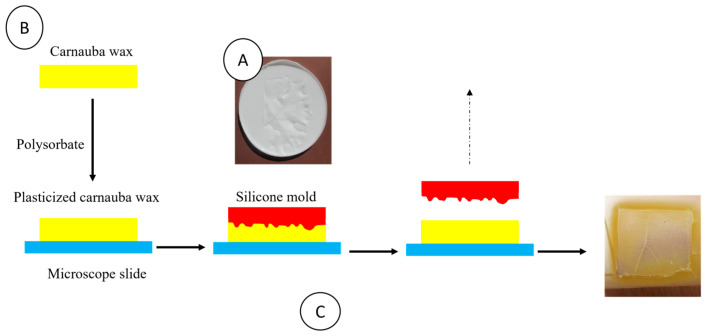
Schematic representation of the described steps of the carnauba wax model fabrication, (**A**) Leaf topography negative printing on silicone, (**B**) Plasticized carnauba wax preparation and (**C**) Leaf topography printing on plasticized carnauba wax.

**Table 1 ijms-26-01031-t001:** Characteristics of the selected chitosan: deacetylation degree (DD by NMR), size (Mn, Ð, SEC-MALS), XPS, and elemental analysis.

DD (%)	M_n_ (kDa) *	Ð	XPS (Atom %)						Elemental Analysis (%)			
			C_1s_	O_1s_	N_1s_	Ca_2p_	Si_2p_	Cl_2p_	C	H	N	O
93	78.7	2.15	70.89	23.05	4.96	0.78	0.23	0.09	40.70	6.85	7.54	43.25

* dn/dc of 0.20 mL/g.

## Data Availability

The data presented in this study are available on request from the corresponding author.

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
