# Peer review of "Feasibility of Genipin to Evaluate Chitosan Rainfastness for Biopesticide Applications"

_ijms, 2025, doi:10.3390/ijms26031031_

Round 1

Reviewer 1 Report

Comments and Suggestions for Authors

The author assesses the possible use of genipin as a chemical reagent with chitosan to produce fluorescence, which is detected here using conventional fluorescence microscopy techniques. The entire study is meticulous and reliable, but some methods and significance of the study are lacking, and the following questions still need to be considered.

 1.      The reason for choosing Parafilm as the leaf model is not sufficient. Although the chemical properties are somewhat similar, the topological structure on the surface of plant leaves is difficult to truly simulate.

2.      What is the principle that genipin can enhance the fluorescence of chitosan? This method is extremely complicated, requiring multiple drops of genipin and drying at 40 degrees for a long time. Then, how to apply this technology in winter or when the temperature is low?

3.       What are the application scenarios of this technology? The author describes using it to evaluate the retention of chitosan after rain wash, but it is difficult to imagine how it can be applied in cropland?

Reviewer 2 Report

Comments and Suggestions for Authors

The manuscript by Meynaud et al. titled “Feasibility of genipin to evaluate chitosan rainfastness for biopesticide applications” reports a study on the quantification of residual chitosan after deposition on leaves and exposure to rainfall mimicking conditions. The study is potentially interesting, but, according to the Reviewer’s opinion, is lacking sufficient originality and innovation compared to the existing literature. The use of genipin to convey fluorescence to chitosan coatings has already been reported in several studies and, in its present state, the manuscript does not provide major advancements on existing challenges, e.g., quantitative and reliable determination of chitosan or improvement on chitosan retention on leaves after rain exposure. The reported results on real vine leaves obtained with MacroFluo highlighted several limitations (extremely high intensity variability, non-linear relationship between intensity and concentration). Hence, the Authors decided to move to model leaves and confocal microscopy. However, the Authors have not compared MacroFluo and confocal microscopy results on model leaves, which could have been useful to understand the limitations of both the leaves model and the quantification techniques.

Round 2

Reviewer 1 Report

Comments and Suggestions for Authors

The authors have addressed the questions I raised, I recommend accepting this article. Nevertheless, it is recommended that the author explore the practical applications of the synthesized nanomaterials in future research to enhance the significance and relevance of the experimental findings.

Reviewer 2 Report

Comments and Suggestions for Authors

The Authors have better explained the contribution of their work to the existing literature and modified the Abstract and Introduction sections accordingly. I still consider this work as very preliminary, so I recommend further future work on real model leaves.